# Choose Your Anchor Wisely:
# Effective Unlearning Diffusion Models
# via Concept Reconditioning

## Abstract

Large-scale conditional diffusion models (DMs) have demonstrated exceptional ability in generating high-quality images from textual descriptions, gaining widespread use across various domains. However, these models also carry the risk of producing harmful, sensitive, or copyrighted content, creating a pressing need to remove such information from their generation capabilities. While retraining from scratch is prohibitively expensive, machine unlearning provides a more efficient solution by selectively removing undesirable knowledge while preserving utility. In this paper, we introduce **COncept REconditioning (CORE)**, a simple yet effective approach for unlearning diffusion models. Similar to some existing approaches, CORE guides the noise predictor conditioned on forget concepts towards an anchor generated from alternative concepts. However, CORE introduces key differences in the choice of anchor and retain loss, which contribute to its enhanced performance. We evaluate the unlearning effectiveness and retainability of CORE on UnlearnCanvas. Extensive experiments demonstrate that CORE surpasses state-of-the-art methods including its close variants and achieves near-perfect performance, especially when we aim to forget multiple concepts. More ablation studies show that CORE's careful selection of the anchor and retain loss is critical to its superior performance.

## 1 Introduction

In recent years, large-scale text-to-image generative models, especially Diffusion Models (DM), have made remarkable advancements in artificial intelligence by exhibiting an unprecedented ability to create high-resolution, high-quality images from text descriptions (Sohl-Dickstein et al., 2015; Ho et al., 2020; Rombach et al., 2022). The versatility and accessibility of diffusion models have led to their widespread adoption across various industries (Croitoru et al., 2023; Kazerouni et al., 2023; Yang & Hong, 2022; Xu et al., 2022).

Despite their broad utility, diffusion models come with inherent risks due to their extensive training on diverse datasets. These models have the potential to generate inappropriate, harmful, or legally sensitive content. For example, Stable Diffusion can produce images that involve pornography, malign stereotypes, and gender and race biases based on the embedded prejudices in their training data, even conditional on non-harmful prompts (Birhane et al., 2021; Schramowski et al., 2023; Larrazabal et al., 2020). They can memorize and reproduce realistic yet inappropriate depictions of individuals without their consent, posing huge privacy risks (Somepalli et al., 2023a,b; Carlini et al., 2023). They can also create misleading or harmful media involving real individuals, such as deepfakes (Mirsky & Lee, 2021). Moreover, they can mimic potentially copyrighted content and replicate styles of real artists, raising legal concerns related to copyright infringement and intellectual

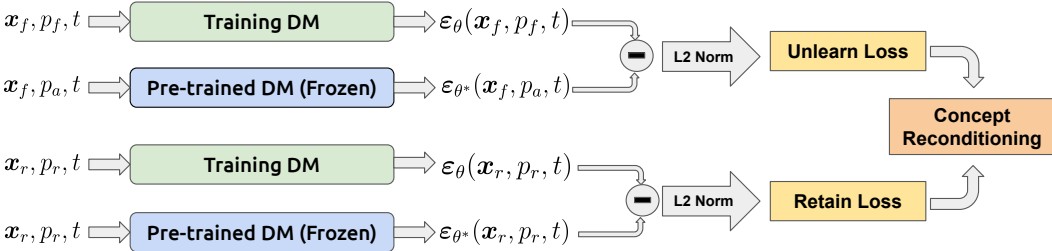

Figure 1: Overview of Concept Reconditioning. $p_f, p_r, p_a$ are the concepts targeted to be forgotten (i.e., *forget concepts*), to be remembered (i.e., *retain concepts*), and to guide unlearning (i.e., *alternative concepts*), respectively. $t$ is the number of steps in the denoising process and is uniformly sampled within $[0, T]$, where $T$ denotes the total number of denoising steps in diffusion models. $\boldsymbol{\varepsilon}_\theta$ is the noise predictor function we aim to optimize, while $\boldsymbol{\varepsilon}_{\theta^*}$ is the noise predictor in the pre-trained diffusion models.

property rights, as well as undermining artistic originality (Shan et al., 2023; Roose, 2022; Liu, 2022; Popli, 2022; Scenario, 2022; Brittan, 2023).

To address these concerns, legislative frameworks such as the European Union's General Data Protection Regulation (GDPR) (Mantelero, 2013; Voigt & Von dem Bussche, 2017) and the US's California Consumer Privacy Act (CCPA) (CCPA, 2018) have established the Right to be Forgotten. These laws mandate that applications must support the deletion of personal information contained in training samples upon user request. Consequently, there is a pressing need for effective methods to mitigate these risks by enabling diffusion models to **unlearn** such undesirable content, ensuring that their deployment is both responsible and aligned with societal values.

A straightforward method is to retrain the model from scratch using a filtered dataset devoid of inappropriate content. However, this approach is computationally intensive and often impractical due to the enormous resources required. For instance, training Stable Diffusion 2.0 on a filtered image set (Schuhmann et al., 2022; Rombach & Esser, 2022) demands approximately 150,000 GPU hours on 256 A100 GPUs. Early attempts to unlearn large-scale generative models include decoding-time guidance and post-generation filtering (Rando et al., 2022; Schramowski et al., 2023); however, these methods do not modify the model weights and can be easily bypassed during deployment. Recent research has pivoted towards more robust fine-tuning-based unlearning approaches that modify a model's weights to effectively forget specific undesirable elements (Gandikota et al., 2023; Fan et al., 2023; Heng & Soh, 2024; Kumari et al., 2023; Wu et al., 2024; Zhang et al., 2024a; Wu & Harandi, 2024; Li et al., 2024b). These methods aim to steer the noise predictor in diffusion models away from the target concepts intended to be forgotten by efficiently fine-tuning a small fraction of parameters.

In this work, we propose **COncept REconditioning (CORE)**, a novel, simple, but effective unlearning method for diffusion model. This method leverages a fixed, non-trainable noise to guide the unlearning process, circumventing the need for dual noise predictors or the use of Gaussian noise as a target. CORE specifically alters the noise prediction mechanism for the target images conditioned on concepts in the forget set (i.e., *forget concepts*), aligning them closer to concepts in the retain set (i.e., *retain concepts*), thereby blurring the distinction between correctly generated images from forget concepts and incorrectly generated ones from retain concepts. We position CORE within a more general framework of *Concept Erasing*, and compare our method with other baselines that fit into this framework. Despite its simplicity, we demonstrate its superiority over existing methodologies through rigorous testing on the UnlearnCanvas framework, and show CORE excels in overall performance including unlearning ability, retainability, and generalization ability, especially when we aim to forget multiple concepts.

Our contributions are summarized as follows.

- We introduce **COncept REconditioning (CORE)** as a new efficient and effective unlearning method on diffusion models, and position it in a broader conceptual framework of concept erasing.

- Extensive empirical validations on UnlearnCanvas showcase that CORE significantly outperforms existing baselines, achieving nearly perfect scores and setting new state-of-the-arts for the overall performance in unlearning diffusion models on UnlearnCanvas. CORE also shows strong capabilities of generalization in unlearning styles.

76 • Ablation studies highlight the benefits of using a fixed, non-trainable target noise over other
77   methods. Additionally, our findings emphasize the superiority of one-to-one concept reconditioning
78   over other schemes of selecting reconditioning concepts.

## 2 Preliminaries

**Machine Unlearning.** Machine Unlearning (MU) refers to the process of systematically removing the influence of specific data points from a trained machine learning model, ensuring that the model forgets information as if the data points were never included in its training set. In this context, let $\mathcal{D}$ represent the training dataset, and let $\mathcal{D}_f \subset \mathcal{D}$ denote the forget set, the subset of data that needs to be unlearned. The retain set, denoted as $\mathcal{D}_r \subset \mathcal{D}$, is the complement of the forget set. The goal of machine unlearning is to produce a new model that closely approximates the performance of retraining from scratch on $\mathcal{D}_r$ while also ensuring that the model does not retain any knowledge of $\mathcal{D}_f$. Unlearning has traditionally been explored in the context of classification models, where the model aims to either forget the influence of specific classes of data or forget some random samples (Cao & Yang, 2015; Bourtoule et al., 2021). In recent developments, machine unlearning has been extended to large generative models, where the model must unlearn specific objectives to ensure that certain generated outputs, such as sensitive, private, copyrighted, or harmful content, will not be generated.

**Unlearning Diffusion Models.** Diffusion models are a class of generative models that have gained significant attention for their ability to generate high-quality images. They work by transforming data distributions through $T$ forward and reverse steps, gradually adding noise to the data and then learning to reverse this process to generate new samples. Mathematically, this can be described by a series of noisy images $\mathbf{x}_0, \mathbf{x}_1, ..., \mathbf{x}_T \in \mathbb{R}^d$, where $\mathbf{x}_0$ is the original image, and $\mathbf{x}_T$ is the Gaussian noise. Latent Diffusion Model (LDM) (Rombach et al., 2022) first compresses high-dimensional pixel-based data into a low-dimensional latent space using an encoder $\mathcal{E}$. It then simulates the diffusion process on the space of latent variables $\mathbf{z} = \mathcal{E}(\mathbf{x})$ and reconstructs the image through a decoder $\mathcal{D}$. For notational simplicity, we do not differentiate between latent variables and pixel-based data, denoting both as $\mathbf{x}$. In this context, let $\varepsilon_\theta(\mathbf{x}_t, p)$ represent the noise estimator parameterized by $\theta$, where $\mathbf{x}_t$ is the noisy observation at step $t$, and $p$ is a conditioning variable such as a class label or text description. The training objective of latent diffusion models is the mean squared error (MSE) between the predicted noise and the true noise across all diffusion steps, expressed as:

$$\mathcal{L}_{\text{MSE}}(\theta) = \mathbb{E}_{p,t,\varepsilon \sim \mathcal{N}(\mathbf{0},\mathbf{I})} \left[ \|\varepsilon - \varepsilon_\theta(\mathbf{x}_t, p)\|_2^2 \right], \tag{1}$$

where $p$ is sampled from a distribution over all prompts and $t$ is sampled uniformly from $[0, T]$. Given a pre-trained latent diffusion model, the objective of unlearning this diffusion model is to ensure that harmful or sensitive content, such as depictions of nudity or violence, can no longer be produced by the model when prompted with the corresponding text descriptions. The challenge lies in balancing the removal of unwanted generations while preserving the model's ability to generate high-quality, appropriate content for normal prompts. The most common unlearning process in diffusion models involves updating the noise estimator to ensure that harmful concepts associated with $\mathcal{D}_f$ are no longer learned or reinforced during the reverse diffusion process. This form of unlearning, often referred to as "concept erasure", is critical for ensuring the safe deployment of generative models in real-world applications. More details are included in Section 3.2.

## 3 Concept Reconditioning

In this section, we propose **COncept REconditionng (CORE)**, a simple yet effective algorithm for unlearning in diffusion models. Our approach focuses on reconditioning the model's learned representations by substituting concepts from the forget set with selected alternative concepts from the retain set. First, we introduce the objective function and key designs within. Then, we position it within the broader framework of *Concept Erasing* and compare it with similar algorithms in prior works to showcase its advantage.

### 3.1 Proposed Method

**Unlearn objective.** In the context of unlearning in diffusion models, we denote the noise predictor in Latent Diffusion Models by $\varepsilon_\theta(\mathbf{x}_t, p)$, where $\mathbf{x}_t$ is the noisy version of the input image $\mathbf{x}_0$ at time

step $t$ generated during the forward diffusion process, $p$ is the prompt associated with the image (e.g., "A cat in the style of Van Gogh"), and $\theta$ represents the model parameters. We use $\varepsilon_{\theta^*}(\mathbf{x}_t, p)$ and $\theta^*$ to denote the pre-trained diffusion model and its parameter. In CORE, we aim to recondition images from the forget set onto alternative concepts. This is achieved by aligning the noise estimator for images in the forget set, conditioned on their original concepts $p_f \in \mathcal{D}_f$, toward the ground truth noise estimator for the same image but conditioned on an alternative concept $p_a$. Mathematically, the unlearn objective function is formulated as

$$\mathcal{L}_f(\theta) := \mathbb{E}_{(p_f, \mathbf{x}_0) \sim \mathcal{D}_f, p_a \neq p_f, t} \left[ \| \varepsilon_\theta(\mathbf{x}_t, p_f) - \varepsilon_{\theta^*}(\mathbf{x}_t, p_a) \|_2^2 \right], \qquad (2)$$

where the expectation is taken over the concept-image pairs $(p_f, \mathbf{x}_0)$ from the forget set, alternative concepts $p_a$ different from $p_f$, and time steps $t$ uniformly sampled from $[0, T]$. Intuitively, this process effectively weakens the association between the images and their original concepts in the model, steering it away from the initial pre-trained associations.

**Alternative concepts.** A key design choice in CORE is the selection of alternative concepts $p_a$ in equation (2). In the unlearning objective, $p_a$ acts as an anchor concept to recondition images from the forget set onto. Previous works typically use an empty string or a single base concept for $p_a$ consistently across all concepts to be unlearned (Zhang et al., 2024c; Gandikota et al., 2023). In contrast, CORE adopts a different approach by pairing each forget concept $p_f$ with a specific alternative concept $p_a$. Our pairing scheme imposes minimal restrictions: the alternative concept $p_a$ does not necessarily have to come from the retain set; it can even be another forget concept different from $p_f$. In our implementation, when the number of concepts to forget is smaller than the number of retain concepts, we map each forget concept to a unique concept in the retain set, rather than using a single base concept for all forget concepts. Meanwhile, when the retain concepts are limited and there are more concepts to forget, we create a one-to-one mapping among the forget concepts themselves. This means that each forget concept $p_f$ is paired with another forget concept $p_a$ (where $p_a \neq p_f$) to serve as its alternative concept during unlearning. Empirically, we show that this one-to-one mapping strategy significantly outperforms methods that consistently use a base concept or randomly sample alternative concepts at each step.

**Retain objective and the full loss function.** To ensure the model continues generating high-quality images for the retain concepts, we introduce a retain loss to regularize the unlearning process. Traditionally, the retain loss is defined as the Mean Squared Error (MSE) between the noise prediction for the retain set and the Gaussian noise vector used to generate the noisy images, similar to the objective used in training a diffusion model (see equation 1). However, in CORE, rather than fine-tuning the noise predictions to match a Gaussian random vector, we instead align them with those generated by the pre-trained diffusion model itself. Mathematically, the retain objective is defined as

$$\mathcal{L}_r(\theta) := \mathbb{E}_{(p_r, \mathbf{x}_0) \sim \mathcal{D}_r, t} \left[ \| \varepsilon_\theta(\mathbf{x}_t, p_r) - \varepsilon_{\theta^*}(\mathbf{x}_t, p_r) \|_2^2 \right], \qquad (3)$$

where $t$ is uniformly sampled in $[0, T]$ and $(p_r, \mathbf{x}_0)$ are concept-image pairs sampled from the retain set. Using $\varepsilon_{\theta^*}(\mathbf{x}_t, p_r)$ as the target helps ensure the model does not deviate too far from its original capabilities, as it leverages the pre-trained model's learned knowledge. Empirical results (see Section 4) demonstrate that aligning the noise predictions with $\varepsilon_{\theta^*}(\mathbf{x}_t, p_r)$, rather than the Gaussian noise, yields better performance. This improvement arises potentially because using the estimated noise from the pre-trained model reduces variance in the unlearned model and stabilize the training process. Interestingly, this phenomenon, where using estimated signals outperforms true signals, has also been observed in other domains in statistics (Robins et al., 1992; Henmi & Eguchi, 2004; Hitomi et al., 2008; Su et al., 2023).

Finally, the complete loss function in CORE combines both the unlearn and retain objectives:

$$\mathcal{L}(\theta) := \mathcal{L}_f(\theta) + \alpha \cdot \mathcal{L}_r(\theta), \qquad (4)$$

where $\alpha > 0$ controls the regularization strength. Intuitively, CORE ensures that the model is steered away from generating images associated with forget concepts while preserving its overall performance on other concepts.

## 3.2 Rethinking Concept Erasing and Reconditioning

At first glance, our proposed objective might seem similar to existing methods for unlearning in diffusion models, as it also involves steering the error predictor on the forget set while keeping it

unchanged on the retain set. However, under closer scrutiny, Concept Reconditioning introduces several key distinctions that set it apart and enable it to outperform previous approaches. Take a broader view of the framework of unlearning diffusion models: unlearning methods for diffusion models that are based on fine-tuning the error predictor $\varepsilon_\theta(\mathbf{x}, p)$ can generally be categorized into two classes: ❶ Concept Erasing (CE): This method works by shifting the noise prediction network for images corresponding to the forget concepts towards an alternative noise distribution. Intuitively, by doing so, it directly acts on $\varepsilon_\theta(\mathbf{x}_t^f, p_f)$, where $\mathbf{x}_t^f$ is the noisy observation for images in the forget set, and misleads them away. ❷ Image Relabeling (IR): In this approach, alternative images that do not match the forget concepts are selected, and the model is fine-tuned on the forget concepts paired with these mismatched images. The model directly acts on $\varepsilon_\theta(\mathbf{x}_t^r, p_f)$ where $\mathbf{x}_t^r$ is the noisy images constructed from the retain set, and effectively overwrites the old knowledge with new associations, forcing it to forget by learning new, incorrect pairings. Mathematically, these two classes can be formulated as

$$\mathcal{L}_{\mathsf{CE}}(\theta) := \lambda \cdot \mathbb{E}_{(p_f, \mathbf{x}_0) \sim \mathcal{D}_f, t} \left[ \| \varepsilon_\theta(\mathbf{x}_t, p_f) - \mathbf{y}_{\mathsf{CE}} \|_2^2 \right], \tag{5}$$

$$\mathcal{L}_{\mathsf{IR}}(\theta) := \lambda \cdot \mathbb{E}_{p_f \sim \mathcal{D}_f, \mathbf{x}_0 \sim \mathcal{D}_r, t} \left[ \| \varepsilon_\theta(\mathbf{x}_t, p_f) - \mathbf{y}_{\mathsf{IR}} \|_2^2 \right]. \tag{6}$$

Here, $\lambda \in \{\pm 1\}$ controls the direction of the objective function. In the CE method, images are drawn from the forget set, while in IR, images come from the retain set. The **target noises** $\mathbf{y}_{\mathsf{CE}}$ and $\mathbf{y}_{\mathsf{IR}}$ can be either random vectors (e.g., Gaussian or Uniform) or derived from a trainable noise predictor.

Many existing unlearning methods fit within this framework. For example, Heng & Soh (2024) suggests $\lambda = -1$ and $\mathbf{y}_{\mathsf{CE}} \sim \mathcal{N}(\mathbf{0}, \mathbf{I}_d)$ in equation (5) in the unlearning objective, while proposing a surrogate objective with $\lambda = 1$ and $\mathbf{y}_{\mathsf{IR}} \sim \mathcal{N}(\mathbf{0}, \mathbf{I}_d)$ in equation (6). The former corresponds to a gradient ascent loss applied to the pre-training objective on forget concepts, while the surrogate objective simply mirrors the standard training loss applied to the forget concepts with retain images. Fan et al. (2023) takes $\mathbf{y}_{\mathsf{CE}}$ in equation (5) as a trainable noise predictor $\varepsilon_\theta(\mathbf{x}_t, p_a)$ where $p_a \neq p_f$ is an alternative concept coming from the retain set. Wu et al. (2024) also proposes this target noise, as well as suggesting an alternative with $\mathbf{y}_{\mathsf{CE}}$ as a uniformly distributed random vector. Kumari et al. (2023) takes $\mathbf{y}_{\mathsf{IR}}$ in equation (6) to be either a standard Gaussian random vector or the error predictor at the last iterate, evaluated at retain images paired with corresponding retain concepts. Even when the objective function appears divergent from this framework, as seen in Gandikota et al. (2023), it can still be decomposed into a linear combination of objective functions in the framework above (see Appendix C).

Although these prior works often include additional techniques such as weight decay (Heng & Soh, 2024), saliency map (Fan et al., 2023), or even applying a monotonic function to the squared loss (Park et al., 2024), the backbone of their unlearning objectives can be positioned into this simple framework or its simple variants. Our method distinguishes itself from prior approaches by its simplicity. Unlike previous methods, CORE requires no auxiliary techniques, and simply optimizing the objective $\mathcal{L}(\theta)$ in equation (4) achieves state-of-the-art results.

Another key distinction is that CORE uses a fixed, non-trainable noise predictor from the pre-trained diffusion model as the target noise. This fixed anchor provides a clearer target noise compared to a trainable network or a random vector with a fixed distribution (e.g., a uniformly distributed random vector). Let us compare the three types of target noises. With a random vector from a fixed distribution (Kumari et al., 2023; Heng & Soh, 2024), there is no guarantee that this manually designed random vector will effectively disrupt the noise predictor conditioned on the forget concepts. A trainable, non-fixed noise (Fan et al., 2023; Kumari et al., 2023; Wu et al., 2024) is unstable during the unlearning process, particularly when aiming to forget many concepts over a long training period, since this target may drift towards an undesired direction. While methods using trainable target noises include a retain term in their loss function, this retain objective directly influences $\varepsilon_\theta(\mathbf{x}_t^r, p_r)$ but not $\varepsilon_\theta(\mathbf{x}_t^f, p_r)$, where $\mathbf{x}_t^r$ and $\mathbf{x}_t^f$ are noisy observations from the retain and forget sets, respectively. In contrast, CORE's use of a non-trainable target noise ensures that the noise predictor always learns from a reference "incorrect" noise estimator derived from the pre-trained model.

## 4 Experiments

In this section, we show CORE outperforms baselines on UnlearnCanvas (Zhang et al., 2024c).

## 4.1 Experiment Setup

**Dataset and Tasks.** UnlearnCanvas is a high-resolution stylized image dataset designed to evaluate diffusion model unlearning methods (Zhang et al., 2024c). The dataset consists of images across 50 unique styles and 20 distinct objects, with 20 images for each style-object combination. Each image is labeled with both a style and an object, making it particularly well-suited for measuring the unlearning effectiveness and the retainability both within a single domain and across domains. In this paper, we mainly focus on style unlearning within the UnlearnCanvas dataset. We define three

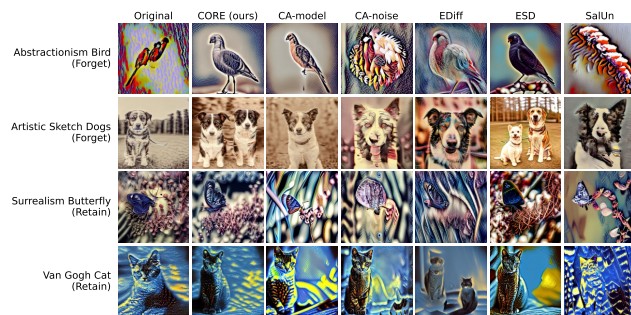

Figure 2: Generated images from the unlearned model. The first column is generated by the fine-tuned Stable Diffusion model before any unlearning. Other columns are generated by the model unlearned by our proposed method and five baseline methods. More images are included in Appendix D.

unlearning tasks, each progressively forgetting more styles: Forget01 (forgetting 1 style), Forget06 (forgetting 6 styles), and Forget25 (forgetting 25 styles).

**Models and Baselines.**

We use a Stable Diffusion v1.5 model (Rombach et al., 2022) to perform the fine-tuning and unlearning, and we also use a vision Transformer (ViT-Large) (Dosovitskiy, 2020) on UnlearnCanvas for style and object classification. Before unlearning the model, the base Stable Diffusion model is fine-tuned on all images from UnlearnCanvas. After completing the unlearning phase, we prompt the unlearned model to generate images conditioned on concepts from both forget and retain sets. The vision Transformer is then used to classify the generated images and calculate the relevant metrics. We compare CORE with several state-of-the-art unlearning methods for diffusion models, including ESD (Gandikota et al., 2023), SalUn (Fan et al., 2023), Ediff (Wu et al., 2024), CA-model and CA-noise (Kumari et al., 2023). See Appendix C for more details.

**Metrics.** Following Zhang et al. (2024c), we use Unlearning Accuracy (UA) to assess the unlearning effectiveness. UA is the percentage of images generated by the unlearned model, conditioned on the forget concepts, which are incorrectly classified by the vision Transformer. A higher UA indicates stronger unlearning capabilities. We measure retainability using two metrics: In-domain Retain Accuracy (IRA) and

| Algorithm | UA (↑) | IRA (↑) | CRA (↑) | SFID (↑) | Total (↑) |
|---|---|---|---|---|---|
| **Forget01** | | | | | |
| Original | 0.00 | 100.00 | 96.67 | 100.00 | 296.67 |
| Ediff | 93.33 | 84.00 | 98.33 | 100.00 | 375.66 |
| CA-model | 96.67 | 80.00 | 92.78 | 100.00 | 369.45 |
| CA-noise | 100.00 | 100.00 | 96.11 | 100.00 | **396.11** |
| SalUn | 53.33 | 98.67 | 92.78 | 95.74 | 340.52 |
| ESD | 100.00 | 66.00 | 96.11 | 96.95 | 359.06 |
| **CORE (ours)** | 93.33 | 98.00 | 96.11 | 100.00 | 387.44 |
| **Forget06** | | | | | |
| Original | 0.00 | 100.00 | 98.33 | 100.00 | 298.33 |
| Ediff | 45.00 | 80.00 | 99.17 | 100.00 | 324.17 |
| CA-model | 85.00 | 81.67 | 88.33 | 88.09 | 343.09 |
| CA-noise | 85.00 | 91.67 | 85.83 | 92.46 | 354.96 |
| SalUn | 90.00 | 83.33 | 98.33 | 88.52 | 360.18 |
| ESD | 100.00 | 75.00 | 100.00 | 93.47 | 368.47 |
| **CORE (ours)** | 90.00 | 100.00 | 97.50 | 99.56 | **387.06** |
| **Forget25** | | | | | |
| Original | 1.20 | 96.54 | 95.29 | 100.00 | 293.03 |
| Ediff | 54.00 | 78.46 | 95.10 | 84.48 | 312.04 |
| CA-model | 68.60 | 78.85 | 95.69 | 81.73 | 324.87 |
| CA-noise | 47.20 | 86.15 | 90.59 | 82.09 | 306.03 |
| SalUn | 51.60 | 77.31 | 87.65 | 82.34 | 298.90 |
| ESD | 90.40 | 46.54 | 99.02 | 88.12 | 324.08 |
| **CORE (ours)** | 91.60 | 95.38 | 97.65 | 100.00 | **384.63** |

Table 1: Performance of CORE and five baseline methods using Stable Diffusion v-1.5 on Forget01, Forget06, and Forget25 in UnlearnCanvas. Unlearning accuracy, In-domain and cross-domain retain accuracy, and scaled FID value serve as main metrics and are summarized in Section 4.1. For details about the scaled FID value, see Appendix B. The best total score is highlighted in **bold**.

---

[1]There are 60 styles in UnlearnCanvas dataset, but in its latest codebase only 50 styles are used. See https://github.com/OPTML-Group/UnlearnCanvas.

Cross-domain Retain Accuracy (CRA). IRA refers to the classification accuracy of generated images prompted with retain concepts, within the same domain (e.g., when forgetting "Van Gogh's style", an in-domain prompt might be "A painting in crayon style"). CRA measures accuracy for retain prompts across domains (e.g., for the same task, a cross-domain prompt might be "A painting of a cat," specifying the object). Additionally, we evaluate the quality of generated images using the scaled FID (SFID) score, which maps the original FID score (Heusel et al., 2017) onto a 0–100 scale, where higher SFID values indicate better generation quality. We also present the summation of all four scores on a scale of 0-100, as a comprehensive measurement of the unlearning capacity and retainability. For more experimental details, see Appendix B.

## 4.2 Results

**CORE achieves the best overall performance.** In Table 1, we present the unlearning effectiveness and retainability of CORE compared to five baseline methods across Forget01, Forget06 and Forget25 tasks from UnlearnCanvas. The "Original" row refers to the performance of the pre-trained model without any unlearning. On Forget01, CORE ranks second overall based on the total score. However, in the more challenging tasks Forget06 and Forget25, CORE consistently achieves the highest total score among all methods, with an increasing performance gap over the baseline methods. Notably, CORE is the only method that maintains strong performance as the size of the forget set grows. In the most difficult task, where 25 out of 50 concepts are targeted for forgetting, CORE achieves the highest unlearning accuracy, in-domain retain accuracy, and scaled FID score, while securing the second-best cross-domain retain accuracy. Compared to its close variants, ESD, CORE achieves similar unlearning accuracy but significantly outperforms in retainability, particularly in cross-domain tasks, due to the adoption of an additional retain loss. Compared to baseline methods that use a trainable noise predictor, such as SalUn and CA-model, CORE excels in forgetting more concepts due to the stability of its non-trainable target, which proves more reliable over longer unlearning periods. Figure 2 shows some generated images using CORE and five baseline methods.

**CORE shows better generalization ability in unlearning styles.** We further investigate CORE's ability to generalize in unlearning styles, aiming to verify that CORE can effectively unlearn specific target styles, instead of simply overfitting to the training objects. To assess this, we train the model on only 10 objects for each forget concept and then evaluate the unlearning accuracy on 10 unseen objects. This tests the model's ability to generalize beyond the specific objects used during training. As shown in Table 2, CORE outperforms all baseline methods in terms of generalization ability.

| Algorithm | UA ($\uparrow$) | IRA ($\uparrow$) | CRA ($\uparrow$) | SFID ($\uparrow$) | Total ($\uparrow$) |
|---|---|---|---|---|---|
| Ediff | 36.67 | 81.67 | 92.50 | 100.00 | 310.84 |
| CA-model | 85.00 | 83.33 | 96.67 | 86.67 | 351.67 |
| CA-noise | 81.67 | 91.67 | 87.50 | 87.62 | 348.46 |
| SalUn | 95.00 | 65.00 | 90.83 | 86.37 | 337.20 |
| ESD | 100.00 | 46.67 | 99.17 | 86.11 | 331.95 |
| **CORE (ours)** | 83.33 | 100.00 | 96.67 | 99.67 | **379.67** |

Table 2: Generalization ability of CORE and baseline methods using Stable Diffusion v-1.5 on Forget06 of UnlearnCanvas. Unlearning accuracy, In-domain and cross-domain retain accuracy, and scaled FID value serve as main metrics and are summarized in Section 4.1. The best total score is highlighted in **bold**.

**The role of non-trainable target noise.** A key design choice in CORE is the use of non-trainable target noise from the pre-trained diffusion model in both the unlearn and retain objectives. This is contrary to other approaches that use trainable noise predictors as targets in the unlearn loss and Gaussian noise vectors as targets in the retain loss. To isolate the specific effects of the non-trainable target noise, excluding the influence of auxiliary techniques like saliency maps, we evaluate several variants of CORE: ❶ We replace $\varepsilon_{\theta*}(\mathbf{x}_t, p_a)$ with $\varepsilon_\theta(\mathbf{x}_t, p_a)$ in equation (2), where $\mathbf{x}_t$ are noisy images from the forget set and $p_a$ is the alternative concept. This variant mirrors the backbone of the unlearn loss used in SalUn (Fan et al., 2023). ❷ We replace $\varepsilon_{\theta*}(\mathbf{x}_t, p_a)$ with a Gaussian noise $\varepsilon$ in equation (2) and apply a negative sign to the unlearn loss. This variant follows the gradient ascent-based method, similar to the unlearn loss in CA-noise (Kumari et al., 2023). ❸ We replace $\varepsilon_{\theta*}(\mathbf{x}_t, p_r)$ with a Gaussian noise $\varepsilon$ in equation (3), where $\mathbf{x}_t$ is noisy observations of images from the retain set. This variant is aligned with the retain loss employed in many baseline methods (Heng & Soh, 2024; Kumari et al., 2023; Wu et al., 2024). The results are shown in Table 3.

**Anchor Selection: How do we approach it?** Another key distinction between CORE and other baseline methods lies in how anchors $p_a$ are selected in the unlearning objective (as defined in equation 2). In CORE, each forget concept $p_f$ is paired with a distinct alternative concept. This

| Unlearn Loss | Retain Loss | UA (↑) | IRA (↑) | CRA (↑) | SFID (↑) | Total (↑) |
|---|---|---|---|---|---|---|
| CORE | CORE | 95.00 | 100.00 | 97.08 | 100.00 | **392.08** |
| $\mathbb{E}\|\varepsilon_\theta(\mathbf{x}_t^f, p_f) - \varepsilon_{\theta^*}(\mathbf{x}_t^f, p_a)\|_2^2$ | CORE | 43.33 | 98.33 | 95.00 | 100.00 | 336.66 |
| $-\mathbb{E}\|\varepsilon_\theta(\mathbf{x}_t^f, p_f) - \varepsilon\|_2^2$ | CORE | 85.00 | 61.67 | 60.00 | 79.48 | 286.15 |
| CORE | $\mathbb{E}\|\varepsilon_\theta(\mathbf{x}_t^r, p_r) - \varepsilon\|_2^2$ | 83.33 | 93.33 | 96.67 | 99.92 | 373.25 |

Table 3: Performance of CORE and its variants on the Forget06 task from UnlearnCanvas. In each variant, one component of the loss function remains unchanged, while the non-trainable target noise in the other component is replaced with alternative approaches. Metrics and are summarized in Section 4.1. The best total score is highlighted in **bold**. Here, $\mathbf{x}_t^f$ and $\mathbf{x}_t^r$ are the noisy observations for images in the forget set and retain set, respectively; $p_f, p_a, p_r$ correspond to forget concepts, alternative concepts, and retain concepts, respectively. $\varepsilon$ denotes the standard Gaussian random vector used to generate $\mathbf{x}_t^f$. Here, we pair each forget concept with one distinct retain concept in all experiments above.

| Scheme for reconditioned concepts | UA (↑) | IRA (↑) | CRA (↑) | SFID (↑) | Total (↑) |
|---|---|---|---|---|---|
| Default (one-to-one) | 91.60 | 95.38 | 97.65 | 100.00 | **384.63** |
| One base concept (all-to-one) | 82.40 | `60.00` | 98.33 | 93.06 | 333.79 |
| Five base concepts (five-to-one) | 93.40 | `84.04` | 98.43 | 96.52 | 372.39 |
| Random concept (one-to-all) | `56.60` | 95.77 | 96.96 | 100.00 | 349.33 |
| Random from five concepts (one-to-five) | `56.40` | 95.58 | 97.75 | 99.78 | 349.51 |

Table 4: Comparison of different alternative concept selection schemes. All experiments are done in the Forget25 task from UnlearnCanvas. In CORE (referred to as "Default"), each forget concept is paired one-to-one with a distinct alternative concept. One base concept: all forget concepts are reconditioned onto a single base concept. Five base concepts: forget concepts are grouped into sets of five, with each group reconditioned to one base concept. Random concept: a random alternative concept is selected for each forget concept at every gradient step. Random from five concepts: each forget concept is paired with five alternative concepts, with one randomly sampled at each step. The best total score is highlighted in **bold**. Significant underperforming results are highlighted in green.

contrasts with other methods that recondition all forget concepts to a single base concept or the empty string. To demonstrate the effectiveness of CORE's one-to-one pairing, we compare different selection schemes: One approach involves pairing each forget concept with a set of alternative concepts (or even the entire retain set) and randomly sampling one at each gradient step to recondition the target images. Another approach reconditions images from multiple or even all forget concepts onto a single base concept. As shown in Table 4, CORE's one-to-one reconditioning scheme significantly outperforms these strategies. Specifically, unlearning accuracy declines sharply when forget concepts are paired with multiple alternatives (one-to-all or one-to-five) and a random alternative is sampled at each step. Conversely, the model's stylistic retainability suffers when all forget concepts are reconditioned to just one or a few base concepts.

# 5 Conclusion and Future Directions

In this paper, we introduce COncept REconditioning (CORE), a novel and effective method for unlearning in diffusion models. CORE leverages a non-trainable target noise from the pre-trained diffusion model to guide both the unlearning and retain objectives, thereby avoiding the pitfalls of using trainable noise predictors or random Gaussian noise targets. Through extensive experiments on the UnlearnCanvas dataset, we demonstrate that CORE consistently outperforms state-of-the-art baseline methods in terms of unlearning effectiveness, retainability, and generalization ability, particularly in challenging tasks involving multiple forget concepts. Moreover, we highlight the importance of a one-to-one concept reconditioning scheme, which proves superior to other anchor selection strategies. There are several promising directions for future research. One key area is improving the efficiency of unlearning, particularly when dealing with a large number of forget concepts. Current methods can still be time-consuming when unlearning many concepts simultaneously. Exploring accelerated unlearning methods while maintaining performance is an exciting avenue. Additionally, future work could investigate the robustness of unlearning methods in dynamic environments, where new concepts might continuously be added to the model, requiring continuous updates without retraining from scratch.

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

# A  Related Works

**Malicious Behavior of Diffusion Models.**   Diffusion models have demonstrated impressive capabilities in generating high-quality, efficient text-to-image outputs (Ho et al., 2020; Song et al., 2020; Rombach et al., 2022). However, these large-scale trained models can pose significant privacy and ethical risks. They are capable of memorizing private images and reproducing objectionable content, such as pornography, private personal photos, malign stereotypes, gender and race biases (Schramowski et al., 2023; Larrazabal et al., 2020; Carlini et al., 2023; Somepalli et al., 2023a; Rando et al., 2022). This mainly stems from the contaminated data sources which involves problematic image-text pairs (Birhane et al., 2021). Furthermore, diffusion models can cause potential issues about copyright infringement by mimicking, or even replicatiing the styles of some specific artistic and their copyrighted work (Shan et al., 2023). Reports showed that AI-generated arts can sometimes be published commercially (Liu, 2022; Popli, 2022; Scenario, 2022) and even awarded prizes (Roose, 2022), raising more serious social concerns about intellectual property violations (Brittan, 2023; Somepalli et al., 2023b; Shan et al., 2023).

**Diffusion Model Unlearning.**   The goal of unlearning diffusion models is to eliminate unwanted concepts and their influence on model outputs. Directly retraining a model to remove such concepts is highly resource-intensive and thus inefficient for large diffusion models (Nichol et al., 2021; Rombach et al., 2022; Schuhmann et al., 2022). Recent research has explored more efficient unlearning techniques. One approach focuses on inference-time methods, which attempt to filter or steer the model away from undesirable outputs during generation (Rando et al., 2022; Schramowski et al., 2023). However, these methods are often limited in effectiveness and can be bypassed, particularly in open-source models (SmithMano, 2023). A more robust alternative involves fine-tuning the model's parameters to actively remove undesirable concepts from its learned representations (Zhang et al., 2024a; Li et al., 2024b; Lyu et al., 2024; Heng & Soh, 2023; Vyas et al., 2023; Gandikota et al., 2024). Some methods are similar to ours: Gandikota et al. (2023); Wu et al. (2024) match the denoising network of correct images given a target concept to another distribution. Fan et al. (2023) additionally adds a saliency map to fine-tune only a small fraction of parameters. Heng & Soh (2024) does gradient ascent on the training loss of diffusion models. Kumari et al. (2023) minimizes the distribution mismatch between the target concept and another anchor concept. We will discuss the difference between our algorithm and theirs in more detail in Section 3.2. Effective though, achieving robust unlearning on complex tasks still remains challenging (Zhang et al., 2024c,d). For a comprehensive review of unlearning techniques in generative models, see Liu et al. (2024a).

**Machine Unlearning.**   Machine unlearning has been extensively explored within classification tasks (Cao & Yang, 2015; Bourtoule et al., 2021; Sekhari et al., 2021; Izzo et al., 2021; Thudi et al., 2022) and is now being applied to large generative models. One popular class of unlearning methods stems from Gradient Ascent(GA) (Jang et al., 2022; Yao et al., 2023; Chen & Yang, 2023; Zhang et al., 2024b). More methods include preference optimization (Zhang et al., 2024b; Maini et al., 2024; Park et al., 2024), model-editing (Meng et al., 2022; Mitchell et al., 2022; Eldan & Russinovich, 2023), knowledge negation (Liu et al., 2024b), representation control (Li et al., 2024a), logits difference method (Ji et al., 2024), random labeling, saliency map (Dou et al., 2024; Tian et al., 2024), and in-context unlearning approaches (Pawelczyk et al., 2023), etc. Some other methods are developed for adversarial unlearning or sequential unlearning tasks (Zhang et al., 2024e; Yuan et al., 2024; Gao et al., 2024). These unlearning methods for language models are orthogonal to our proposed method for unlearning diffusion models.

## B  Experiment Details

**Hyperparameter.**  All experiments are done using one 80GB NVIDIA A100 GPU. We use an open-sourced Stable Diffusion v-1.5 for all experiments (Rombach et al., 2022), which is first fine-tuned on all data in UnlearnCanvas before any unlearning process, and the fine-tuned model is provided by Zhang et al. (2024c). As suggested in prior works (Gandikota et al., 2023; Zhang et al., 2024c), we only fine-tune the cross-attention in U-Nets in the Stable Diffusion and freeze all other parameters when doing unlearning. Following Zhang et al. (2024c), we use the first three images for each style and object for training. For CORE, we run 25 epochs in Forget01 and Forget06, and 100 epochs in Forget25. In testing the generalization ability of unlearning styles, where the testing and training objects are distinct, we double the epochs in Forget06. We use Adam with a constant learning rate of $1 \times 10^{-5}$ in CORE, and the batch size is set to 4. We set $\alpha = 1.0$ in equation (4). The hyperparameters used for training the baseline methods are described in Appendix C.

**Scaled FID Values.**  Scaled FID (**SFID**) is a modified version of Fréchet Inception Distance (FID) (Heusel et al., 2017), which ranges from zero to infinity and measures the quality of generated images. A lower FID value indicates a higher generation quality. To measure the overall performance of unlearning algorithms, we convert the original FID value into Scaled FID value, which ranges from 0 to 100 and increases when the generation quality grows. We compute the original FID value for the base model and the unlearned model, denoted as $\mathbf{FID}_0$ and $\mathbf{FID}_M$, respectively. SFID is then defined as

$$\mathrm{SFID}_M = \min\left\{100 \times \frac{\mathbf{FID}_0}{\mathbf{FID}_M}, 100\right\} \tag{7}$$

A model with better retainability tends to have higher SFID values. In our experiments, we compute SFID values on the retain set.

## C  Baseline Methods Overview

In this section, we introduce baseline methods, discuss how they relate to our proposed approach, and describe their training procedures. For the most part, the training setup for these baseline methods follows Zhang et al. (2024c). We set the alternative concept as one common base concept (one base style) in Forget01. For each step, we randomly sample one alternative concept from the retain set in Forget06. In Forget25, we create a bijection from the 25 concepts in the forget set and the other 25 concepts in the retain set. In Forget25, we have also tried to pick a random alternative concept at each step, but this worsens the performance for all baselines by a large margin.

**ESD (Gandikota et al., 2023).**   ESD is the first method that offers both efficiency and effectiveness in unlearning for diffusion models. It utilizes a more complex unlearning objective without incorporating a retain objective. As a result, ESD's retainability is generally outperformed by other methods. The objective function for ESD is defined as follows:

$$\mathcal{L}_{\mathsf{ESD}}(\theta) := \mathbb{E}_{(\mathbf{x}_0, p_f) \sim \mathcal{D}_f, t} \left\| \varepsilon_\theta(\mathbf{x}_t, p_f) - \left( \varepsilon_{\theta^*}(\mathbf{x}_t, p_0) - \eta \big( \varepsilon_{\theta^*}(\mathbf{x}_t, p_f) - \varepsilon_{\theta^*}(\mathbf{x}_t, p_0) \big) \right) \right\|_2^2, \quad (8)$$

where $(\mathbf{x}_0, p_f)$ are sampled from the forget set, $t$ is uniformly sampled from $[0, T]$, $\varepsilon_\theta$ and $\varepsilon_{\theta^*}$ are the current and pre-trained noise predictors in diffusion models. Here, $p_0$ is a base concept, which can be an empty string (Gandikota et al., 2023) or a base style in UnlearnCanvas. In our experiments, according to Gandikota et al. (2023), we set $\eta = 1.0$, batch size to 1, and the learning rate to $1 \times 10^{-5}$, and we run 1000 gradient steps.

Although the objective in ESD seems to be very different from our framework of concept erasing, we can still fit it into our framework in Section 3.2 via proper decomposition. Namely,

$$\begin{aligned}
\mathcal{L}_{\mathsf{ESD}}(\theta) :&= \mathbb{E}_{(\mathbf{x}_0, p_f) \sim \mathcal{D}_f, t} \left\| \varepsilon_\theta(\mathbf{x}_t, p_f) - \left( \varepsilon_{\theta^*}(\mathbf{x}_t, p_0) - \eta \big( \varepsilon_{\theta^*}(\mathbf{x}_t, p_f) - \varepsilon_{\theta^*}(\mathbf{x}_t, p_0) \big) \right) \right\|_2^2 \\
&= \mathbb{E}_{(\mathbf{x}_0, p_f) \sim \mathcal{D}_f, t} \left\| \varepsilon_\theta(\mathbf{x}_t, p_f) - (1 + \eta)\varepsilon_{\theta^*}(\mathbf{x}_t, p_0) + \eta \varepsilon_{\theta^*}(\mathbf{x}_t, p_f) \right\|_2^2 \\
&= \mathbb{E}_{(\mathbf{x}_0, p_f) \sim \mathcal{D}_f, t} \Big( \varepsilon_\theta(\mathbf{x}_t, p_f)^2 + (1 + \eta)^2 \cdot \varepsilon_{\theta^*}(\mathbf{x}_t, p_0)^2 + \eta^2 \cdot \varepsilon_{\theta^*}(\mathbf{x}_t, p_f)^2 \\
&\quad + 2\eta \cdot \varepsilon_\theta(\mathbf{x}_t, p_f) \cdot \varepsilon_{\theta^*}(\mathbf{x}_t, p_f) - 2(1 + \eta) \cdot \varepsilon_\theta(\mathbf{x}_t, p_f) \cdot \varepsilon_{\theta^*}(\mathbf{x}_t, p_0) \\
&\quad - 2\eta(1 + \eta) \cdot \varepsilon_{\theta^*}(\mathbf{x}_t, p_0) \cdot \varepsilon_{\theta^*}(\mathbf{x}_t, p_f) \Big) \\
&= \mathbb{E}_{(\mathbf{x}_0, p_f) \sim \mathcal{D}_f, t} \Big( (1 + \eta) \cdot \big( \varepsilon_\theta(\mathbf{x}_t, p_f) - \varepsilon_{\theta^*}(\mathbf{x}_t, p_0) \big)^2 - \eta \cdot \big( \varepsilon_\theta(\mathbf{x}_t, p_f) - \varepsilon_{\theta^*}(\mathbf{x}_t, p_f) \big)^2 \\
&\quad + \eta(1 + \eta) \cdot \big( \varepsilon_{\theta^*}(\mathbf{x}_t, p_0) - \varepsilon_{\theta^*}(\mathbf{x}_t, p_f) \big)^2 \Big) \\
&= \underbrace{(1 + \eta)\mathbb{E}_{(\mathbf{x}_0, p_f) \sim \mathcal{D}_f, t} \big( \varepsilon_\theta(\mathbf{x}_t, p_f) - \varepsilon_{\theta^*}(\mathbf{x}_t, p_0) \big)^2}_{(a)} \\
&\quad \underbrace{- \eta \mathbb{E}_{(\mathbf{x}_0, p_f) \sim \mathcal{D}_f, t} \big( \varepsilon_\theta(\mathbf{x}_t, p_f) - \varepsilon_{\theta^*}(\mathbf{x}_t, p_f) \big)^2}_{(b)} \\
&\quad + \underbrace{\eta(1 + \eta)\mathbb{E}_{(\mathbf{x}_0, p_f) \sim \mathcal{D}_f, t} \big( \varepsilon_{\theta^*}(\mathbf{x}_t, p_0) - \varepsilon_{\theta^*}(\mathbf{x}_t, p_f) \big)^2}_{(c)}.
\end{aligned}$$

Since term (c) in the last line is a constant independent of $\theta$, we can omit it in the loss function. The remaining two terms (a) and (b) can both fit into the Concept Erasing framework (see equation 5). Term (a) is equivalent to choosing $\lambda = (1 + \eta)$ and $\mathbf{y}_{\mathsf{CE}} = \varepsilon_{\theta^*}(\mathbf{x}_t, p_0)$, while term (b) is equivalent to choosing $\lambda = -\eta$ and $\mathbf{y}_{\mathsf{CE}} = \varepsilon_{\theta^*}(\mathbf{x}_t, p_f)$.

**SalUn (Fan et al., 2023).**   Saliency Unlearning (SalUn) introduces a saliency mask to the diffusion model parameters before unlearning. This mask, based on the absolute gradient scale for the forget concept, identifies the most important parameter subsets for unlearning targeted concepts, enabling efficient unlearning that edits only a small portion of the model. The loss function for SalUn is given

by:

$$\mathcal{L}_{\mathsf{SalUn}}(\theta) := \underbrace{\mathbb{E}_{(\mathbf{x}_0, p_f) \sim \mathcal{D}_f, t, p_r \neq p_f} \left\| \varepsilon_\theta(\mathbf{x}_t, p_f) - \varepsilon_\theta(\mathbf{x}_t, p_r) \right\|_2^2}_{\text{unlearn objective}} + \underbrace{\beta \cdot \mathbb{E}_{(\mathbf{x}_0, p_r) \sim \mathcal{D}_r, t, \varepsilon} \left\| \varepsilon - \varepsilon_\theta(\mathbf{x}_t, p_r) \right\|_2^2}_{\text{retain objective}},$$

(9)

where $\varepsilon$ is the standard Gaussian random vector used to generate $\mathbf{x}_t$, and $p_f$ and $p_r$ are forget concepts and retain concepts, respectively. $t$ is sampled uniformly from $[0, T]$. In contrast to CORE, which uses $\varepsilon_{\theta*}$ as the target for the retain objectives, SalUn uses the Gaussian random vector $\varepsilon$. Their unlearn objective can fit in the framework in equation (5) with a trainable network as the target noise. This can lead to target degradation during the unlearning process, especially when multiple concepts need to be unlearned. Following Fan et al. (2023) and Zhang et al. (2024c), we take $\beta = 1.0$. We use a learning rate of $1 \times 10^{-5}$ and a batch size of $4$. We run 10 epochs in Forget01 and 100 epochs in Forget06 and Forget25.

**EDiff (Wu et al., 2024).** EraseDiff (EDiff) formulates the objective as follows:

$$\mathcal{L}_{\mathsf{EDiff}}(\theta) := \underbrace{\mathbb{E}_{(\mathbf{x}_0, p_f) \sim \mathcal{D}_f, t, \varepsilon_f} \left\| \varepsilon_\theta(\mathbf{x}_t, p_f) - \varepsilon_f \right\|_2^2}_{\text{unlearn objective}} + \underbrace{\beta \cdot \mathbb{E}_{(\mathbf{x}_0, p_r) \sim \mathcal{D}_r, t, \varepsilon} \left\| \varepsilon - \varepsilon_\theta(\mathbf{x}_t, p_r) \right\|_2^2}_{\text{retain objective}}. \quad (10)$$

The retain objective is similar to that in SalUn, but the unlearn objective differs. Here, $\varepsilon_f$ is a uniformly distributed random vector, which serves as the target noise. This unlearn objective aligns with the concept erasing framework (equation 5), where $\mathbf{y}_{\mathsf{CE}}$ is uniformly distributed. EraseDiff simplifies the diffusion process by solving it as a first-order optimization problem, reducing computational complexity. In our experiments, we use a batch size of $4$ and a learning rate of $5 \times 10^{-5}$. We run 5 epochs in Forget01 and 50 epochs in Forget06 and Forget25.

**CA (Kumari et al., 2023).** Concept Ablation (CA) matches the image distribution from the forget set to an anchor concept. They design two objective functions: a model-based one and a noise-based one. The model-based CA objective is defined as

$$\mathcal{L}_{\mathsf{CA-model}}(\theta) := \underbrace{\mathbb{E}_{(\mathbf{x}_0, p_f) \sim \mathcal{D}_f, t} \left[ \omega_t \left\| \varepsilon_\theta(\mathbf{x}_t, p_f) - \varepsilon_\theta(\mathbf{x}_t, p_0).\mathsf{sg}() \right\|_2 \right]}_{\text{unlearn objective}}$$
$$+ \underbrace{\lambda \cdot \mathbb{E}_{(\mathbf{x}_0, p_r) \sim \mathcal{D}_r, t, \varepsilon} \left\| \varepsilon - \varepsilon_\theta(\mathbf{x}_t, p_r) \right\|_2^2}_{\text{retain objective}}. \quad (11)$$

Here, $\omega_t$ is a time-dependent weight applied to the loss, $p_0$ is a fixed base concept from the retain set, and .sg() denotes the stop-gradient operator. The noise-based objective is defined as

$$\mathcal{L}_{\mathsf{CA-noise}}(\theta) := \underbrace{\mathbb{E}_{(\mathbf{x}_0, p_f) \sim \mathcal{D}_f, t, \varepsilon} \left[ \omega_t \left\| \varepsilon_\theta(\mathbf{x}_t, p_f) - \varepsilon \right\|_2 \right]}_{\text{unlearn objective}}$$
$$+ \underbrace{\lambda \cdot \mathbb{E}_{(\mathbf{x}_0, p_r) \sim \mathcal{D}_r, t, \varepsilon} \left\| \varepsilon - \varepsilon_\theta(\mathbf{x}_t, p_r) \right\|_2^2}_{\text{retain objective}}. \quad (12)$$

In both objectives, $\varepsilon$ is the standard Gaussian random vector used to generate $\mathbf{x}_t$. In our experiments, we use a batch size of $4$ and a learning rate of $1.6 \times 10^{-5}$. We run 200 gradient steps in Forget01 and 100 epochs in Forget06 and Forget25.

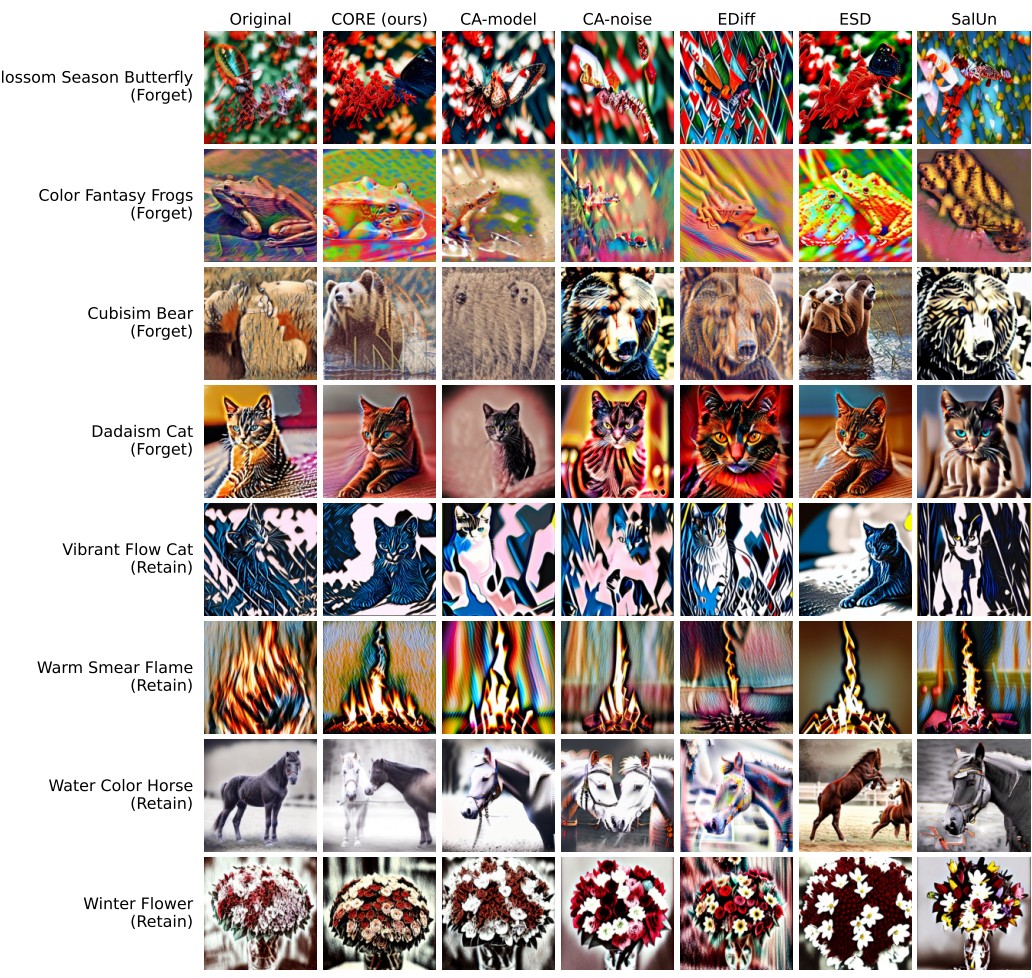

Figure 3: Additional generated images from the unlearned model. The first column is generated by the fine-tuned Stable Diffusion model before any unlearning. Other columns are generated by the model unlearned by our proposed method and five baseline methods.

# D   More results

In this section, we present more images generated from our experiments on UnlearnCanvas in Figure 3.