# OpenReview forum: "Choose Your Anchor Wisely: Effective Unlearning Diffusion Models via Concept Reconditioning"
_NeurIPS.cc/2024/Workshop/SafeGenAi — SafeGenAi Poster_

### Official Review · Reviewer_W5Ti · 2024-10-09
**The paper provides effective unlearning by designing a different denoising loss**

**Rating:** 7
**Confidence:** 4

**Review:**

**Strengths**

* The method CORE uses noise predicted by a pre-trained model to compute unlearn and retian loss, differentiating it from previous methods within the same framework.
* The experiments of erasing multiple concepts on UnlearnCanvas dataset demonstrate the effectiveness of their proposed method.
* The ablation study on different variants of CORE provides a better understanding of each loss component.

**Weaknesses**

* The paper can consider modifying Figure 1 to visualize the differences between their method and other similar ones, e.g., by plotting illustrations of other methods' objectives and highlighting their distinctions. Also, the abstract can be further revised to clearly emphasize the novelty of CORE, which is currently missing.
* It would be more convincing if the paper provided a principled explanation for using a noise predictor from the pre-trained model as the target.
* The paper claims its method is efficient, but there is no evaluation of its efficiency compared with others.
* For future revision, the paper could add detailed analyses of catastrophic retaining and unlearning rebound for sequential unlearning, and add experiments on unlearning multiple styles or objects simultaneously.

---

### Official Review · Reviewer_GZyW · 2024-10-09
**Simple and intuitive approach to unlearning with good empirical proof-of-concept**

**Rating:** 7
**Confidence:** 3

**Review:**

**Summary**

The authors introduce "concept reconditioning" as an approach to "unlearn" knowledge in diffusion models, or remove information while preserving utility. They propose a novel objective function for unlearning, where the noise estimator in the diffusion model is updated to align with noise associated with an "alternative" concept, some concept unrelated to the concept to be forgotten. Empirical evaluations on the UnlearnCanvas benchmark validate improvements by the proposed approach over other unlearning baselines. Ablations and sensitivity analyses validate the loss formulation and alternative concept selection strategy.

**Strengths**
* Preliminaries were clearly explained, enough for me to mostly understand the intuition behind concept erasing. While I have working knowledge of diffusion models and the general formulation, it's not my #1 area.
* The method is fairly intuitive and easy to understand. The design choices are fairly logical, and additional explanation is provided in Section 3.

**Weaknesses**
* I'm struggling to get why the objective in Eq. 2 helps with unlearning: it's not totally clear to me why conditioning images from the forget set onto alternative concepts is appropriate. Doesn't this mean that the model is still able to generate images from the "forget" concept, so long as one has knowledge of the alternative concept(s)? Especially if the alternative concept is drawn from the retain set, couldn't someone prompting a model to generate something in the retain set inadvertently surface a "forget set" concept? If the goal of the technique is to prevent *intentional* generation of images from undesired concepts, I could understand, but I'm not sure that's aligned with the framing as-is.
* How is the one-to-one mapping between forget concepts and alternative concepts created? Is it an arbitrary permutation? Or is there a specific pairing algorithm used?
* I don't quite get the 2nd experiment in Section 4.2 (better generalization ability in unlearning styles). Isn't CORE already unlearning specific target styles?

**Other comments**
* CORE intuitively seems like a variation of image relabeling. To that end, it seems like Eq. 4 can be reformulated as "minimize
$\mathcal{L}\_r(\theta)$
subject to $\mathcal{L}\_f(\theta)$ bounded above." (i.e., simply switching which term is the regularizer by dividing the objective by a constant $1/\alpha$). Given this formulation, the objective  resembles a *constrained* version of Eq. 6 ($\mathcal{L}\_{IR}(\theta)$) where $\mathbf{y}\_{IR} := \varepsilon\_\theta(x_t, p_a)$, for some randomly sampled concept in the retain set $p_a \sim \mathcal{D}\_r$.  I wonder if highlighting this connection could help contextualize the benefit of CORE w.r.t. the unlearning literature; e.g., CORE improves upon naive optimization of $\mathcal{L}\_{IR}$ by 1) finding a more clever distribution of $y\_{IR}$ than, say, $\mathcal{N}(0, \mathbf{I})$, while 2) staying close to the original diffusion model. This is discussed to some extent in L210-222 but could be made clearer. Is this a fair assessment of the method?
* In Table 1, reporting unlearning accuracy on the original data seems a little odd for an unlearning evaluation — aren't we simply testing the ability of the model to generate images consistent/without the concept of interest (as judged by the ViT classifier)? Would it make more sense to report the *delta* of other methods w.r.t. "Original?"
* Also, I don't understand the utility of adding all the metrics to summarize in Table 1. The proposed approach seems to be competitive or better than the baselines across all metrics, **and the gap widens as the # of concepts to forget increases.** I would highlight that trend more clearly — that's independently interesting.
* I suggest adding a row to Table 3, where the retain loss is dropped completely. That would test whether we can get unlearning "for free" (w/o an explicit retain loss). In my opinion, that is highly doubtful\ but that experiment would tell us how "sharp" the tradeoff is between unlearning and retaining the original DM's performance.
* The sensitivity analysis of alternative concept selection strategy in Table 4 is really interesting. I wonder if there's some theoretical results that can be shown that explains 1) why mapping multiple forget concepts to one alternative concept underperforms (and the problem seems to worsen as each base concept has more "responsibility" for a larger set of forget concepts), and 2) why mapping one concept to multiple concepts isn't helpful for unlearning. The latter is quite counterintuitive (i.e., aren't we spreading the probability mass for the "forget" concept more widely) to me.

---

### Official Review · Reviewer_SaWQ · 2024-10-09
**Paper proposed an unlearning method revolving around class reconditioning. Great writing and insights!**

**Rating:** 7
**Confidence:** 3

**Review:**

The writing was very detailed and well-written. It was a joy to read. Though effective, a major concern lies in the use of retaining data $\mathcal{D}_r$. Due to major privacy concerns, $\mathcal{D}_r$ might not be available, which might break the proposed method. How would the authors put forward or reformulate this?

The experiments were carefully designed and \textbf{CORE} achieved SOTA results. The appendix gives additional experimental details. However, \textbf{CORE} could also be checked on other datasets and contrasted against Selective Amnesia [1]


[1] Alvin Heng and Harold Soh. Selective amnesia: A continual learning approach to forgetting in deep 399 generative models. Advances in Neural Information Processing Systems, 36, 2024.

---

### Official Review · Reviewer_Ekti · 2024-10-12
**concept reconditioning in machine unlearning**

**Rating:** 8
**Confidence:** 5

**Review:**

This paper proposes a novel approach for machine unlearning in diffusion models, focusing mostly on concept erasing. The proposed method is simple, but seems to be quite effective, comparing to state-of-the-art methods (like SalUn).

The method is introduced in a clear way, and the paper itself is well-written. CORE achieves good performance on a set of experimental tasks, which is an obvious strength of this paper. Finally, I think that this work should be interesting for the SafeGenAi workshop audience.